# Stable Gaze Tracking with Filtering Based on Internet of Things

**DOI:** 10.3390/s22093131

**Published:** 2022-04-20

**Authors:** Peng Xiao, Jie Wu, Yu Wang, Jiannan Chi, Zhiliang Wang

**Affiliations:** 1School of Computer and Communication Engineering, University of Science and Technology Beijing, Beijing 100083, China; xp_0311@163.com (P.X.); wzl@ustb.edu.cn (Z.W.); 2School of Automation and Electronic Engineering, University of Science and Technology Beijing, Beijing 100083, China; s20200543@xs.ustb.edu.cn (J.W.); wangyu5@comac.cc (Y.W.)

**Keywords:** gaze tracking, Kalman filtering, pupil tracking

## Abstract

Gaze tracking is basic research in the era of the Internet of Things. This study attempts to improve the performance of gaze tracking in an active infrared source gaze-tracking system. Owing to unavoidable noise interference, the estimated points of regard (PORs) tend to fluctuate within a certain range. To reduce the fluctuation range and obtain more stable results, we introduced a Kalman filter (KF) to filter the gaze parameters. Considering that the effect of filtering is relevant to the motion state of the gaze, we design the measurement noise that varies with the speed of the gaze. In addition, we used a correlation filter-based tracking method to quickly locate the pupil, instead of the detection method. Experiments indicated that the variance of the estimation error decreased by 73.83%, the size of the extracted pupil image decreased by 93.75%, and the extraction speed increased by 1.84 times. We also comprehensively discussed the advantages and disadvantages of the proposed method, which provides a reference for related research. It must be pointed out that the proposed algorithm can also be adopted in any eye camera-based gaze tracker.

## 1. Introduction

Gaze is a significantly crucial communication human behavioral cue, which reveals a lot of information, such as internal thoughts and mental states. Gaze tracking is a technology that adopts visual detection methods to obtain a user’s current PORs. It is widely used in various applications, such as human–computer interaction [1,2,3], virtual reality research [4,5,6], healthcare/clinical assessment and diagnosis [7,8], and action recognition [9,10]. These applications are also important foundations for building an Internet of Things society and a Web of Things society. In this paper, we attempt to improve the performance of gaze tracking in an active infrared source gaze-tracking system.

Several non-invasive vision-based gaze-estimation methods have been proposed in the last few decades. Generally, these methods can be grouped into model-based and appearance-based methods [11]. Appearance-based methods generally require a single camera to capture the appearance of the human eye and regress the gaze from the appearance [12]. It usually requires three components [13]: an effective feature extractor to extract the gaze features from high-dimensional raw image data; a robust regression function to learn the mappings from appearance to human gaze; and a large number of training samples to learn the regression function. Because no specific equipment is required, including its excellent robustness against the environment, the research on appearance-based methods is becoming increasingly popular [1,2,13]. However, it is faced with two major challenges: the computation cost is large, and its estimation accuracy is limited. Model-based methods rely on eye feature extraction (such as eye corners or iris center localization) to learn a geometric model of the eye, and then infer gaze direction using these features. They can be further divided into 2D and 3D methods. Three-dimensional methods construct a geometric 3D eye model and estimate gaze directions based on the model, while 2D methods directly adopt the detected geometric eye features, such as pupil center and glint, to regress the point of gaze. Model-based methods can obtain high-precision eye pictures using special equipment, which increases the modeling accuracy. However, the robustness is often unsatisfactory (for example, sensitive to head movement). In our research, we adopted a pair of eye-tracking glasses to capture eye images and construct a 2D gaze-estimation model.

The accuracy of pupil positioning is critical for the general performance of the system. In the pupil detection process, various interference factors influence pupil tracking, such as the reflection point of the infrared illuminator on the cornea, uneven illumination, and glass occlusion. To improve the accuracy of pupil tracking, significant efforts have been made [6,9,14,15]. In our experiments, we noticed that when we gaze at a fixed point, the calculated POR is often unstable, and it fluctuates within a certain range. This is believed to be caused by pupil-tracking errors. Specifically, because the experimental environment cannot be kept perfectly consistent (e.g., the head may move slightly), the eye pictures obtained by the camera may be slightly different, thus resulting in possible changes in the extracted line-of-sight parameters. Such small differences are amplified by the mapping model and they eventually lead to significant fluctuations in the estimated PORs. The most practical solution to minimize fluctuations is filtering the line-of-sight parameters. Considering that tracking is a time sequence problem, we choose the classic Kalman filter (KF) method [16], which owns excellent performance and fast speed. To optimize KF, we design that the measurement noise varies with the speed of gaze. The experiment indicates that the variance of the POR error decreased from 147.04 ${\mathrm{pixel}}^{2}$ to 38.48 ${\mathrm{pixel}}^{2}$.

Real-time is also an important evaluation metric for gaze tracking. Although several studies [17,18,19] have significantly improved its accuracy, its speed is slowed down by complex models and huge computations. We are committed to improving the speed of gaze tracking in daily environments. Pupil tracking is a necessary step in eliminating background interference and extracting pupil parameters. To quickly locate the pupil area from the original image, we propose the replacement of the conventional eye detection method with the current popular correlation filtering tracking method, and then adopt a high-precision pupil-fitting method to extract the line-of-sight parameters. Compared with eye detection, the speed of the pupil tracking increased by 1.84 times, and the image area of the pupil area was reduced by 93.75%. In addition, we solely utilized the left and right pupil margin points to circumvent the effect of eyelid occlusion on the pupil parameter extraction.

The contributions of our work are summarized as follows:(1)To speed up the process of line-of-sight parameter extraction, we replace the conventional eye detection method with the correlation filtering-based pupil-tracking method, and use the star-ray method to reduce the points used in ellipse fitting.(2)To reduce the fluctuation of the gaze PORs, we introduce a Kalman filter to filter the pupil–cornea reflection vector, and achieve excellent results. To further optimize KF, we design the measurement noise that varies with the speed of gaze.(3)We present a comprehensive analysis of the proposed method in this paper, pointing out its advantages, disadvantages, and applicable scenarios, which provides a practical reference base for similar studies.

The remainder of this paper is organized as follows. Section 2 introduces some related studies. Section 3 presents the overall architecture of the method in this study and details its improved parts. In Section 4, the experimental results verify the effectiveness of the proposed method. In Section 5, we provide a comprehensive summary of this study, and a future research direction is presented.

## 2. Related Work

In this section, we briefly summarize the relevant literatures on gaze-estimation models and eye parameter extraction methods.

### 2.1. Gaze-Estimation Model

Several researchers have proposed novel model-based gaze methods. Chi et al. [17] estimated the 3D gaze using the real-time calculated transformation matrix represented by the kappa angle, which mitigates the inherent errors triggered by the optical axis reconstruction in conventional methods and simplifies the algorithm for gaze estimation. Liu et al. [20] proposed an iris feature-based 3D gaze-estimation method, using a single camera and a single light source. The 3D line-of-sight is estimated from the optical axis and positional relationship between the optical and visual axes, and then optimized using a binocular stereo vision model. These methods have high accuracy because the eye model can simulate the human eyeball structure precisely. At the same time, they usually rely on person-dependent calibration, which make it difficult to build a general model for every people. Appearance-based approaches aim to find the mapping function from image appearance to gaze direction or gaze location. The mapping function is usually learned using different machine learning algorithms such as K-Nearest Neighbor [21], Random Forest [22], or Artificial Neural Networks [23,24,25]. A current review of appearance-based model can be found in [26,27]. Early appearance-based methods usually rely on hand craft features, and it is difficult to maintain high accuracy in different settings. Recently, the fast-developing CNN with strong representational power makes it the most popular method to estimate human gaze. Kim et al. [23] published near infrared dataset NVGaze, which contains big amount of collected real images and high-resolution synthetic images with detailed parameters. He et al. [24] proposed a lite CNN model without face input and a few shot calibration schemes, which further increases accuracy. Guo et al. [25] proposed a Tolerant and Talented training scheme to get better accuracy and robustness. Although appearance-based methods do not require any knowledge about human vision and they only need simple eye-detection techniques, these methods require a large amount of data to learn the mapping function. This limits their applications in real-world applications. In this study, we propose another method to improve the estimation performance by eliminating noise in the line-of-sight parameters.

### 2.2. Eye Parameters Extraction

For the model-based gaze estimation method, the extraction of eye parameters is a crucial step, and it can be divided into two parts: eye local image segmentation and eye-parameter extraction. The most commonly used eye features are the pupil center, pupil ellipse, iris ellipse, or track iridial features. Nasro et al. [28] gives a detailed summary of the pupil-detection methods. Despite several recent advancements in eye-tracking technology [29], three factors continue to adversely impact the performance of eye-tracking algorithms: (1) reflections from the surroundings and from intervening optics; (2) occlusions due to eyelashes, eyelid shape, or camera placement; and (3) small shifts in the eye-tracker position caused by slippage [30]. Biswas et al. [31] presented a method for iris detection and recognition, which comprises several steps: segmentation, normalization, feature encoding, feature extraction, and classification. Hough transform was adopted in detecting the pupil center of an iris. Based on the detected edges, PuRe [32] develops a specific edge segment selection that wipes off the major edges that do not belong to the pupil contour and a conditional segment combination scheme to select the best edge. Li et al. [33] proposed to combine the eyeball geometry to eliminate many outliers of pupil contours. After calibrating the eyeball center with several continuous frames, pupil contours are constrained by the projections of the real pupil. Recently, the CNN algorithm [34] has been used to detect pupil centers. It will have better accuracy than existing model-based methods if it uses a large amount of training data. However, due to the high variability of pupil images in real-world environments, building an efficient database is challenging. Besides, since this algorithm treats pupil detection as a black box, the internal principles cannot be understood, making it difficult to generalize to solve similar problems, such as iris detection. Based on the fast radial symmetry transform, FREDA [35] seeks to find the pupil center as the highest radial symmetry in the image. In addition, pupil tracking [36] is employed to predict the pupil position in the next frame based on the present and previous frames, but it is unsuitable for detecting a single pupil image. Considering the high speed of the tracking method, we attempt to utilize a correlation filter [37] to quickly locate the pupil area, and then extract the high-precision pupil parameters using elliptical fits.

## 3. Proposed Method

This section provides a detailed description of the proposed gaze-estimation framework. Firstly, an overview of the framework is provided in Section 3.1. Then, we introduce eye-parameter detection in Section 3.2. Subsequently, we elucidate pupil tracking using a kernelized correlation filter (KCF) [37] in Section 3.3. Finally, we present gaze estimation with filtered eye parameters in Section 3.4.

### 3.1. Overall Architecture

As illustrated in Figure 1, a complete 2D mapping-based gaze-estimation process can be divided into three parts: calibration, eye parameter extraction, and POR estimation. The calibration (in the red dashed box) determines the mapping relationship between the pupil and cornea reflection vector and the PORs on the screen. In this study, we adopted a nonlinear mapping model, based on second-order polynomial fitting.

Eye parameter extraction (in the green dashed box) refers to the process of obtaining eye parameters from original images, such as the pupil center coordinate and the inclination angle of the fitting ellipse. Eye-parameter extraction is generally performed by first detecting the eye region image, and then extracting the eye parameters. Compared to the conventional method, we introduced two alterations. One alteration is to extract the image near the pupil using a tracking method, instead of extracting the image near the eye using a detection method. The other is to fit the pupil ellipse with the edge points on the left and right sides of the pupil, instead of all edge points. Based on the high speed of tracking, we adopted a tracking method to extract smaller pupil images. For 2D gaze estimation, we need the coordinates of the pupil center; hence, it is perfectly possible to directly construct the pupil corneal reflection vector from the tracking results. However, the accuracy of the tracking usually cannot meet the requirements of gaze estimation; hence, tracking can solely be used to quickly locate the pupil, and then the pupil center must be determined with a high-precision ellipse fitting. In addition, considering that the pupil may be occluded when blinking, this study adopts the star-ray method to obtain the edge points on the left and right sides of the pupil for fitting, which effectively circumvents the influence of occlusion.

Gaze estimation (in the gold dashed box) is the process of estimating the PORs on the screen, according to the eye parameters. In the 2D gaze-estimation model, we usually take the pupil and Purkin spot centers to form a pupil-corneal reflection vector, which can be substituted into the pre-trained polynomial model to obtain the PORs. In actual experiments, even if we stare at a fixed point, the final estimated PORs will not be static, but would fluctuate within a certain range. Sometimes, PORs may even deviate significantly, which adversely influences the overall estimation effect. To reduce the magnitude of fluctuations of PORs and improve the accuracy of gaze estimation, we filtered the eye parameters with a Kalman filter before putting them into the mapping model to calculate the PORs.

### 3.2. Eye Parameters Extraction

Eye parameters include the parameters of the pupil and Purkin spot. The shape of the pupil is approximately an ellipse; hence, its parameters generally correspond with the parameters of the fitted ellipse. The parameters of the Purkin spots generally refer to the center coordinates because of their small size. In this study, we extracted the pupil and Purkin centers to form the pupil–corneal reflection vector. Figure 2 illustrates this process. First, we used a KCF tracker to obtain the local pupil image from the original input image captured by the camera. The tracking process is comprehensively described in the next section. Then, we extracted the pupil and Purkin spot center coordinates from the pupil image. Regarding the pupil center, the pupil image is binarized with a low threshold, and the incomplete area is complemented by morphological filtering. The star-ray method was adopted to select edge points in the angle regions of [−70°, 60°] and [120°, 250°] for ellipse fitting. For the Purkin spot center, a higher threshold was adopted to segment the Purkin spot area from the pupil image, and the center was obtained by the centroid method. Finally, the pupil and Purkin centers constitute the pupil–corneal reflex vector required by the mapping model.

### 3.3. Pupil Tracking Using KCF

Pupil tracking is part of the eye parameter extraction in our framework (see Figure 1), which is used to quickly locate the pupil region. Based on practical considerations, we chose the KCF. The flowchart of pupil tracking is shown in Figure 3.

Principle: The principle of KCF tracking is to determine the correlation between the filter and search area, and the position of the maximal response is the position of the target. The objective function of a KCF tracker is denoted as a ridge regression,
(1)$$\underset{w}{\min}\|Xw-y{\|}_{2}^{2}+\lambda \|w{\|}_{2}^{2},$$
where λ , X, y, and w represent the weight coefficient of the regular term, circulant matrix form of the input image, standard response, and weight coefficient of the inputs, respectively. Because λ is usually defined as a constant, the correlation filter w is the only parameter that is required to be calculated and updated.

To solve nonlinear problems, a kernel trick is introduced. w is expressed as a linear combination of weights α and samples in feature space φ(x), $w=\underset{i}{\sum }\alpha_{i}\varphi \left(x_{i}\right)$. Hence, the variables under optimization are α, instead of w. The kernel function κ is defined as $\kappa \left(x,x^{\prime }\right)=\varphi^{T}\left(x\right)\varphi^{T}\left(x^{\prime }\right)$. The dot products between all pairs of samples form an n×n kernel matrix K. The solution to the kernelized version of the ridge regression is given by [38], $\;\alpha ={\left(K+\lambda I\right)}^{-1}y$. If K is a circulant matrix, the solution can be simplified as
(2)$$\hat{\alpha }=\frac{\hat{y}}{\hat{k}+\lambda },$$
where k represents the first row of the kernel matrix K and ^ denotes the discrete Fourier transform (DFT). Different kernels lead to different expressions of α, and the most used kernels are Gaussian and linear kernels. For a Gaussian kernel, k can be calculated as
(3)$$k=\exp\left(-\frac{1}{\sigma^{2}}\left({\left\|x\right\|}^{2}+{\left\|x^{\prime }\right\|}^{2}-2F^{-1}\left(\hat{x}⊙{\hat{x^{\prime }}}^{*}\right)\right)\right),$$

More equations for the generation vector of the kernel matrix can be found in [37]. Finally, the regression function for all candidate patches is computed using $\;f\left(z\right)={\left(K^{z}\right)}^{T}\alpha$.

Initialization: Correlation filter initialization calculates the initial correlation filter $\;{\hat{\alpha }}_{0}={\hat{y}}_{0}/\left({\hat{k}}_{0}+\lambda \right)$, and sets a few other parameters, such as the size of the search window. The position of the tracking target in the first frame must be determined using a detection method. In this study, we employed the classic Haar feature-based cascade classifier method in OpenCV. To ensure that the pupil image contains Purkin spots, we expanded the detected pupil bounding box by 1.25 times. Figure 4 presents some results of the initialization.

Estimation: By convolving the search window with the learned correlation filter, we obtain a response map. The maximum value in the map was the most impossible position of the target. The formula is given by
(4)$$z_{k}=\max\left({ℱ}^{-1}\left({\hat{k}}_{k}⊙{\hat{\alpha }}_{k}\right)\right),$$

Convolution operations usually incur high computational costs; hence, the speed is significantly slow. Fortunately, KCF skillfully changes the convolution operation in the time domain into multiplication operations in the frequency domain, and substantially decreases the computation.

Updating: The filter ${\hat{\alpha }}_{k}$ can be updated by
(5)$$\hat{\alpha }=\left(1-\eta \right)\hat{\alpha }+\eta \hat{\alpha_{k}},$$
where η∈(0,1) is the learning rate.

### 3.4. Gaze Estimation with Filtered Parameters

The conventional estimation method simply needs to substitute the eye parameters into the mapping model to obtain the PORs. In this study, the Kalman filter step was introduced to reduce the signal fluctuation.

The Kalman filtering algorithm is a method for estimating the minimum variance error of the state sequence of a dynamic system, and it has a wide range of applications in target tracking, data smoothing, etc. The state change of the target at two adjacent moments can be denoted by the state equation:(6)$$x_{k}=A_{k}x_{k-1}+B_{k}u_{k}+q_{k},$$
where state $x_{k}\in R^{n}$, $A_{k}$, and $B_{k}$, represent the state vector, an n×n state-transition matrix, and an n×1 control–input matrix, respectively. $u_{k},\;q_{k}∼N\left(0,Q_{k}\right)$, and $Q_{k}$ denote the control-input vector, process noise, and process noise covariance matrix, respectively. In addition, the intrinsic relationship between state $x_{k}$ and measurement $\;z_{k}\in R^{m}$ can be denoted by the measurement equation:(7)$$z_{k}=H_{k}x_{k}+r_{k},$$
where $H_{k}$ is an m×n state-to-measurement transformation matrix, $r_{k}∼N\left(0,R_{k}\right)$ is the measurement noise, and $R_{k}$ represents the measurement noise covariance matrix.

In the tracking problem, the state equation is usually modeled as a constant-velocity model or a constant-acceleration model. They have a similar process. KF tracking can be summarized into two steps: prediction and correction, using the five equations expressed below. A detailed proof can be found in [16]. In our method, the filtered parameter is the pupil-corneal reflection vector, so the form of the state vector $x_{k}$ is $\left[x,y,v_{x},v_{y}\right]$, where x and y are horizontal coordinate and vertical coordinate, respectively, and $v_{x}$ and $v_{y}$ are the velocity in corresponding direction. The form of measurement $z_{k}$ is set to [x,y]. For convenience, $A_{k}$, $H_{k}$, $Q_{k}$ are set to constants, and $B_{k}$ and $u_{k}$ are set to zero.

In the prediction step, we calculate the predictions for state ${\bar{x}}_{k}$ and its error covariance ${\bar{P}}_{k}$ at time k. The predicted state ${\tilde{x}}_{k}$ can be calculated by Equation (6) without the noise term,
(8)$${\bar{x}}_{k}=Ax_{k-1}.$$

The predicted error covariance ${\bar{P}}_{k}$ can be calculated by Equation (9),
(9)$${\bar{P}}_{k}=AP_{k-1}A^{T}+Q.$$

In correction step, we first calculate the Kalman gain $K_{k}$,
(10)$$K_{k}={\bar{P}}_{k}H^{T}{\left(H{\bar{P}}_{k}H^{T}+R_{k}\right)}^{-1}.$$

Then the corrected state ${\hat{x}}_{k}$ is obtained by weighted sum of predicted state ${\bar{x}}_{k}$ and measurement $z_{k}$,
(11)$${\hat{x}}_{k}={\bar{x}}_{k}+K_{k}\left(z_{k}-H{\bar{x}}_{k}\right).$$

The error covariance is $P_{k}$ updated by,
(12)$$P_{k}=\left(1-K_{k}H\right){\bar{P}}_{k}.$$

If we know the initial state $x_{0}$, initial error covariance $P_{0}$, and measurement sequence $z_{1:k}$, state $x_{k}$ can be estimated by iterating Equations (8)–(12). In fact, even if the measurement $z_{k}$ is unknown, we can still estimate $x_{k}$ by assuming that the measurement equals the prediction, accordingly, $z_{k}=H{\bar{x}}_{k}$. This implies $x_{k}={\bar{x}}_{k}$, and the process noise is zero in this time interval.

In our experiment, we test both the constant-velocity model and the constant-acceleration model. The state-transition matrix A for a constant-velocity KF filter is usually set to
(13)$$A_{CV}=\left[\begin{array}{cccc} 1 & 0 & \Delta T & 0\\ 0 & 1 & 0 & \Delta T\\ 0 & 0 & 1 & 0\\ 0 & 0 & 0 & 1 \end{array}\right].$$

A for a constant-acceleration KF filter is usually set to,
(14)$$A_{CA}=\left[\begin{array}{cccc} 1 & 0 & \Delta T & {\left(\Delta T\right)}^{2}/2\\ 0 & 1 & 0 & \Delta T\\ 0 & 0 & 1 & 0\\ 0 & 0 & 0 & 1 \end{array}\right].$$

ΔT is the interval time, and can be set to 1 in frame-to-frame tracking. The state-to-measurement transformation matrix H is set to same form in two kinds of filter,
(15)$$H=\left[\begin{array}{cccc} 1 & 0 & 1 & 0\\ 0 & 1 & 0 & 1 \end{array}\right].$$

The process noise covariance matrix $Q_{k}$ and the measurement noise covariance matrix $R_{k}$ are both 4×4 diagonal matrices, and reflect the reliability of the predicted state ${\bar{x}}_{k}$ and the measurement $z_{k}$, respectively. A reasonable assumption is that the prediction ${\bar{x}}_{k}$ should be trusted more when the gaze is still, while the measurement $z_{k}$ should be trust more when the gaze moves. The faster the gaze moves, the more the measurement should be trusted. $Q_{k}$ is determined by the model accuracy, and is taken to be constant. $R_{k}$ changes according to the estimated velocity of gaze point.
(16)$$R_{k}=\{\begin{array}{cc} w\tilde{R}+\left(1-w\right)R_{k} & if\;\tilde{R}<R_{k-1}\\ \tilde{R} & else \end{array}\;,$$
where $\tilde{R}$ is a parameter that varies with speed,
(17)$$\tilde{R}=R_{max}\cdot max\left\{v_{k-L},\ldots ,v_{k}\right\}\;,$$
and $R_{max}$ is a predefined maximum covariance for the measurement, which is bigger than Q.

Now, a key point is how to calculate the gaze velocity. For the convenience of description, the time in the formula below uses superscript t. The most direct method for estimating gaze velocity is eye difference [39]. The amount of eye movement was computed using subsequent frames.
(18)$$m^{t}=\sqrt{\overset{D_{e}}{\underset{n=i}{\sum }}{\left(\psi_{i}^{t}-\psi_{i}^{t-1}\right)}^{2}},$$
where $\psi_{i}^{t}$ is the ith pixel of the cropped images at time *t*, and $D_{e}$ denotes the number of pixels in the cropped images. The relative amount of movement can be calculated by mapping m to the range [0,1], using a logistic function.
(19)$$S_{m}^{t}=\frac{1}{1+e^{-\beta_{m}\left(m^{t}-\alpha_{m}\right)}},$$
where the condition $\alpha_{m}>0,\;\beta_{m}>0$ controls the shape of the logistic function. To obtain the gaze velocity at time t, $S_{m}^{t}$ is multiplied by the measured difference in gaze points.
(20)$$v^{t}=S_{m}^{t}\left(q^{t}-q^{t-1}\right),$$
where $q={\left[x,y\right]}^{T}$ denotes the gaze point.

Finally, the coordinates of the pupil center and those of the Purkin spot constitute the reflection vector, which can be substituted into the mapping model to calculate the POR.

## 4. Experiments and Discussion

### 4.1. Experimental Environment

The head-mounted gaze tracking system used in our experiment consists of a laptop and a pair of eye-tracking glasses, as illustrated in Figure 5. The laptop (see Figure 5a) is used to show the target to be gazed. The eye-tracking glasses (see Figure 5b) comprise a front-view camera and two sets of left and right light-ring modules. The light-ring module comprises eight infrared LED lights and an eye camera (see Figure 5c). The glasses are connected to the computer via a USB interface, and can continuously transmit the wearer’s eye images to the computer, while the computer completes the image processing and algorithm calculations. The image obtained by the system is a single eye image of the subject at a close distance. The camera can capture up to 60 RGB images with a size of 640 × 480 pixels per second. The screen is 9.3 inches with resolution of 1280 × 720. The CPU of the laptop used for the experiment was an Intel (R) Core (TM) i7-4710MQ with 2.20 GHz frequency.

### 4.2. Pupil Tracking

In this section, we compare the results of detection and tracking in the extraction of pupil images under various conditions. Although many novel correlation filtering methods have been proposed, more complex models usually require slower tracking speeds and higher hardware requirements. Considering that the purpose of tracking is to quickly locate the pupil area, we prioritized fast tracking algorithms such as MOSSE [40] and KCF [37]. Simultaneously, we also compared the tracking performance of different features. The experiment was divided into two groups: the cases when the eyeballs moved randomly with normal blinking (group A), and the cases when the eyeballs moved randomly with occasional squinting (group B). The parameters of each tracking algorithm were the default parameters in the original papers. As a reference, we adopted the results of the detection as a baseline.

The pupil extraction results are presented in Figure 6. In group A, MOSSE (red box), KCF-gray (gold box), and KCF-hog (green box) can all follow the pupil well. The speed of the human blink is 250 ms each time. Generally, the probability that the experimental camera captures a completely closed eye image is substantially low, and most of the captured eye images are half-closed, as illustrated in the fourth picture. At this point, the upper part of the pupil is blocked by the eyelids, and the Purkin spots above are also lost. It can be observed that although the tracking accuracy was reduced, it still remained within an acceptable range. If the eyeball rotation angle is excessively large, it may trigger irregular imaging of the light source on the sclera, as illustrated in the fifth picture. In addition, the gray feature tends to be more suitable for pupil tracking than the hog feature.

In group B, we observed a similar phenomenon; however, because the pupil was blocked more often, the tracking accuracy was worse than that in group A. In addition, because the detection method requires pupil fitting, when the pupil is blocked, it is difficult to guarantee the fitting accuracy. In the fourth image of groups A and B, the pupils are severely occluded, and the tracking and detection accuracies are significantly reduced.

To observe the tracking effect of the three tracking methods more intuitively, we repeated the experiment of group A 10 times, and then plotted a precision graph, as illustrated in Figure 7. It can be observed that KCF-gray performs better than MOSSE and KCF-hog. The width × height of the pupil in the original image is approximately 72 × 68. Considering that the template size is generally 1.5–2.5 times the target size, even if the tracking result deviates from the detection result by 15 pixels, the tracking result can still completely contain the pupil area image and satisfy the experimental requirements.

Detailed tracking values are presented in Table 1. The average speeds of MOSSE, KCF-gray, and KCF-hog are {652, 426, 439} FPS, with average errors of {9.16, 7.24, 7.63} pixels, respectively. Although the speed of the KCF method is slightly slower than that of the MOSSE method, the accuracy of the KCF method is higher. Considering this fact, we chose KCF-gray as the pupil-tracking method in this study. The classic Haar feature-based cascade classifier method detects the eyes at a speed of 232 frames per second. In contrast, the pupil detection speed of the tracking method proposed in this study increased by 1.84 times. Although the classic method is already very fast under our current experiment equipment, faster pupil detection can leave more space for follow-up research, such as using more accurate and more complex gaze modes.

Note that a large area of black pixels exists in the upper left corner of the original image (see Figure 2), which adversely affects threshold segmentation. The pupil image obtained by pupil tracking solely contains images near the pupil, which cannot only directly eliminate a large amount of background noise interference, but can also reduce the size of the image to be processed to 6.25% of the original image.

### 4.3. Gaze Estimation

In this section, we evaluate the impact of Kalman filtering on the gaze-estimation results. We designed two experiments on different tasks. In the first task, the target remained still (gaze task), and in the second task, the target moved (saccade task). In each task, we compared three POR results: results with no filtering (raw), filtering with a constant-velocity Kalman filter (kal_cv), and filtering with a constant-acceleration Kalman filter (kal_ca). The target and the calibration points are controlled by a program, and we number the 9 calibration points as 1–9 for convenience, as shown in Figure 8.

#### 4.3.1. Gaze Task

In the gaze task, the subject stared at each calibration point successively. We processed the images collected at different points separately, calculated the POR of each image, and finally drew nine POR figures (see Figure 9). Here, we select the results of calibration points 4 (P4) and 8 (P8) for a detailed discussion, as illustrated in Figure 10.

All nine POR figures exhibit the same pattern: the centers of the green, red, and blue circles are very close; however, their radii increase successively. In fact, the POR figure of P8 exhibits the closest radius for the green and red circles. Nevertheless, the radius of the green circle remains smaller than that of the red circle, and the green circle in the other figures is significantly smaller than that of the red circle.

The close proximity of the circle centers indicates that the mean errors of the three methods are similar. The detailed mean errors at the nine calibration points are presented in Table 2. In general, kal_cv appears to have the smallest mean error, while raw exhibit the largest mean error, although the difference is negligible. Even the raw method achieved the best results at P2. Considering that these disparities depict the pixel distances on the screen, they are not representative; i.e., the filtering has a negligible effect on improving the accuracy of the gaze estimation.

The radius of the circle depicts the distribution range of the PORs. According to the radius, filtering can reduce the fluctuation range of the PORs, and the kal_cv model is better than the kal_ca model. The middle and right columns of Figure 10 present the x- and y-values of the POR, respectively. The blue line in the figure represents the unfiltered result (raw), and the green and red lines depict the results of kal_cv and kal_ca, respectively. In the first frame, the results for the three lines were the same, because at the initial moment, we set the state value of KF to be equal to the observed value. Evidently, the filtered curve is smoother, which indicates that filtering reduces the sharpness of the POR fluctuations; i.e., the variance of the filtered results is smaller. It should be noted that the variance here reflects the variation of the POR coordinates, and not the variation of the error, although they are essentially the same. The variance of the error at each calibration point is presented in Table 3. The results of kal_cv and kal_ca is significantly smaller than those of raw, and kal_cv achieved the best results.

For further comparison, we also add a 3D method [17] as basic method to evaluate the effect of filtering. Except for the gaze model, other settings are the same. The mean and variance of the errors at all points are presented in Table 4. In 2D model group, the means of the three methods {raw, kal_cv, kal_ca} are {55.56, 54.34, 54.78} pixels, and their variances are {147.04, 38.48, 72.88} pixel^2^, respectively. There is almost no change in the mean error, and the best result (kal_cv) is only 3.96% less than raw. Compared with raw, the variances of kal_cv and kal_ca are significantly reduced by 73.83% and 50.44%, respectively. In the 3D model group, we observed the same phenomenon. The mean error of {kal_cv, kal_ca} decreases by {1.58%, 1.05%}, while the variance of them decreases by {69.84%, 51.09%}. No matter what the gaze model is, KF will decrease the mean error little, while decreasing the variance significantly. Furthermore, filtering improves the mean and variance less due to the higher accuracy of the 3D model.

In summary, when we stare at a fixed point, filtering barely improves the estimation accuracy; however, it can significantly reduce the error variance, such that kal_cv performs better than kal_ca.

#### 4.3.2. Saccade Task

We designed the following experiments for the saccade task. A gaze point moves on the screen according to a predetermined trajectory, as shown in Figure 11. The subject watches the gaze point, and the camera records a series of eye photographs. During the experiment, the head remained still, and the gaze point moved with constant horizontal and vertical velocities of 20 and 10 pixels per second, respectively, and then stays for 3 s at each turning point. Note that Figure 11 is only a schematic diagram of the movement trajectory of the gaze point, and only one spot was visible at a time, and the spot had the same color. The nine red points indicate the locations of the nine standard calibration points, which are used to calculate the map model in the calibration step. According to the motion state of gaze, we divided the entire process into 19 time periods marked as T1–T19, as shown in Table 5.

The estimated results for the sight point are presented in Figure 12. When the gaze point is located at 10 turning points, the estimated POR coordinates remain nearly unchanged, because the point remains stationary, and the x- and y-values exhibit a horizontal trend. When the gaze point moves between the turning points, the PORs also move, and at least x or y exhibits a slanted trend. Because few eye images are collected in one position when the gaze point is moving, it is unsuitable for quantitative error analysis. Therefore, we primarily analyze the results when the gaze point is stationary.

From the general effect of Figure 12, the filtered result maintains the same trend as the raw result. The right column depicts the results in the T7 time period when the gaze point is at point P8. The only difference in the comparison with the gaze task is that the fluctuations of filtered x are greater during the starting period of T7. When the gaze point moves from P6 to P4 through P8, x initially decreases, remains constant, and then decreases. When the gaze point is suddenly stationary, the Kalman filter calculates the predicted state (see Equation (8) with the former state (x decreases), and this prediction is significantly smaller than the raw value, which eventually leads to a smaller estimated KF result than the raw value. This is illustrated in the graph as a large trough in x with filtering. When the gaze point suddenly moves from P8 to P4, the KF model predicts with the former state (x remains constant), which leads to a larger estimation.

In addition, the trough of kal_ca is higher than that of kal_cv, and it approaches raw faster. This is because kal_ca adopts a constant acceleration model that can adapt faster to state changes. However, this also causes the variance of the kal_ca method to be greater than kal_cv when it is stationary. Similarly, we can analyze the y-value plots for T7. The KF model makes predictions based on the previous state (y increases), which leads to a large final estimation. As illustrated in Figure 12, a large peak exists in the KF model.

We also selected four representative subplots for analysis (see Figure 13). In general, the trend of the filtered y-values and unfiltered y-values remain the same; however, the filtered y-value is smoother, and its peak and valley are closer to the standard position (blue line), which indicates that Kalman filtering facilitates the reduction of fluctuations. Figure 13a presents the results of y values at time period T1, when the gaze point is at P1, and it exhibits the same characteristics as those in the gaze task. The starting points of the lines in Figure 13a are initially the same, but later differ from each other. Figure 13b shows the results of the y-values at time period T5, when the light source is at P8. As can be observed, both lines start with relatively large fluctuations, and then they both stabilize within a certain range. This is because the estimated error when the gaze point is moving is greater than that when the gaze point is stationary; hence, when the gaze point changes from moving to stationary, the error undergoes a process from large to small; however, it quickly approaches the stationary estimated error. Figure 11d and Figure 13c present the POR estimation results of the gaze point at the same location (P2), at different times (T3 and T11). The first time of the gaze point trajectory is P1P2P6; hence, the y-value initially remains unchanged before increasing. The second time is P4P2P3, where the y-value initially decreases, remains unchanged, and then increases. From P1 to P2, the y-value is constant, and the error does not change significantly (see Figure 13c). From P4 to P2, the y-value decreases; hence, a large error exists at the initial moment of T11 (see Figure 13d).

From the above, the state of the gaze point at the last time period significantly impacts the prediction of the current moment, which is also the reason for the significant peaks and valleys. We conducted further experiments to explore this effect. We repeated the process of moving the gaze point from P6 to P4 through P8 several times, varying the speed of the gaze point and the dwell time at the turning points to ensure that the same number of images were obtained when the gaze point was motionless, including different numbers of images when the gaze point was moving. Figure 14 presents the estimated y-values of P8 with different numbers of motion images between P6 and P8. We took 50 motionless images, and 20, 60, and 100 motion images in Figure 14a, Figure 14b, and Figure 14c, respectively.

It can be observed that as N increases, the peaks of the green and red lines in the initial stage become smaller and converge to the blue line faster. This is because as N increases, the state difference between two adjacent frames decreases, and the estimated value of Equation (8) becomes more accurate. It is conceivable that when N is sufficiently large, the image in Figure 14 resembles the image from the GAZE experiment.

We also intercepted the y-value plot when the gaze point was moving, as illustrated in Figure 15. The filtered curve (green and red lines) is smoother than the unfiltered curve (blue line), and as N increases, the filtered curve is more approximate to the unfiltered curve. This is the same pattern as when the gaze point is stationary.

Finally, we tracked a randomly moving gaze point, and the results are presented in Figure 16. The black, blue, green, and red lines represent the trajectory of the gaze point, POR without filtering, POR filtered by kal_cv, and POR filtered by kal_ca, respectively. It can be observed that the filtered curve is smoother. Among them, the green line is smoother than the red line; however, its deviation may be greater. This indicates that kal_ca can adapt better to state changes than kal_cv.

### 4.4. Discussion

To evaluate the effect of the improvements proposed in Section 3, we conducted two experiments: pupil tracking and gaze estimation.

In the pupil tracking experiment, we adopted tracking instead of detection to quickly determine the pupil area image. It is important to note that it is difficult to accurately determine the center of the pupil. Therefore, accurate pupil-parameter detection is required in the follow-up. This is because in target tracking, the target image is usually a rectangular region image, and the image center is the target center. Regarding the pupil, its shape is determined and approximated to that of an ellipse. By fitting the ellipse, we can obtain the precise parameters of the pupil. If a rectangular area is used instead of an elliptical area, the background point is actually regarded as a part of the target; hence, the tracking accuracy is certainly worse than detection. In addition, if the pupil scale change and tilt angle are considered, the scaling and rotation of the target template must be considered when tracking, which will trigger additional computations, and the speed advantage of tracking may not be reflected. Therefore, we utilized the superior correlation filtering tracking method instead of the conventional deep learning tracking method. Pupil tracking experiments indicate that the tracking method can effectively extract the pupil region images, and its speed is faster than that of the detection method. In addition, the effect of the gray feature is better than that of the hog feature, which may be owing to the fact that the gray value distribution of pixels in pupil region is more significant than that of gradient distribution.

In the gaze-estimation experiment, we compared the PORs, with and without filtering. First, we conducted a gaze experiment to evaluate the effect of filtering on line-of-sight estimation, and then we conducted a saccade experiment to evaluate the robustness of the filtering against the movement of the gaze point. The experimental results indicate that filtering can effectively reduce fluctuations in the line of sight. When the gaze point is stationary, the kal_cv model works better, and when the gaze point is in motion, the kal_ca model is more suitable. Via experiments, we observed the limitations of Kalman filtering: when the gaze point moves, especially when it moves randomly, the error may become larger. This occurs because the movement model established by the Kalman filter does not adequately describe the actual movement of the gaze point. The principle of filtering is to modify the current results according to existing data points. The proposed prediction models are meaningful, and the corrections are plausible only when all the data conform to the same distribution pattern. Unlike the movement of objects in nature, the actual movement of the pupil is very complicated, which makes it almost impossible for us to model the extracted pupil parameters. In this experiment, we adopted the Kalman filter with an idealized constant velocity model and a constant acceleration model. Although optimal results have been obtained, there is still room for further improvement, such as considering the state transition matrix, measurement noise, and system noise changes over time.

## 5. Conclusions

Gaze estimation is a means to interpret the semantic information of eye pictures. This study focused on improving the speed and accuracy of two-dimensional gaze estimation, and proposed two improvements: adopting the correlation filter tracking method to extract the image of the pupil area, and employing the Kalman filter to filter the line-of-sight parameters.

To quickly extract the image of the pupil area, this study adopted the superior KCF tracking method instead of the conventional target detection method. Although it is difficult to accurately locate the center of the pupil using the tracking method, it is sufficient for extracting images of the pupil area. Compared with eye detection, the speed of pupil tracking can be increased by 1.84 times, and the determined pupil area is 93.75% smaller than the detected eye area image. This does not only eliminate a large number of pixels in the non-pupil region, but also reduces computation required for the subsequent extraction of pupil parameters.

To reduce the fluctuation of the POR estimation error, a Kalman filter was proposed to filter the line-of-sight parameters. The experimental results demonstrated that filtering can significantly reduce the fluctuation range of the POR. When the gaze point was stationary, the KF based on the constant velocity model (kcf_cv) reduced the variance of the error by 73.83%, which is better than the KF based on the constant acceleration model (kcf_ca). When the gaze point was in motion, although the error fluctuation of kcf_ca was still greater than that of kcf_cv, its accuracy was higher. In addition, when the motion state of the gaze point changed abruptly, the filtering produced a large error, and the kcf_cv error was greater than that of kcf_ca.

The method proposed in this study has a certain reference significance for improving the performance of the gaze-estimation system. The improvement is reflected in reducing the POR fluctuation of the basic method, thereby improving tracking stability, while it does not help to improve the accuracy. Although originally proposed for 2D gaze estimation, it is applicable to 3D gaze estimation or other mapping models. Gaze tracking is valuable research, and faces many challenges and difficulties, such as occlusion, head movement, appearance of objects. Most of the current research solve problems in constrained local spaces and cannot be generalized to the open world. Recently, the powerful representation capabilities of deep neural networks may bring us a feasible way. In the future, we will explore combing filtering with deep neural networks to build an accurate and stable tracking system.

## Figures and Tables

**Figure 1 sensors-22-03131-f001:**
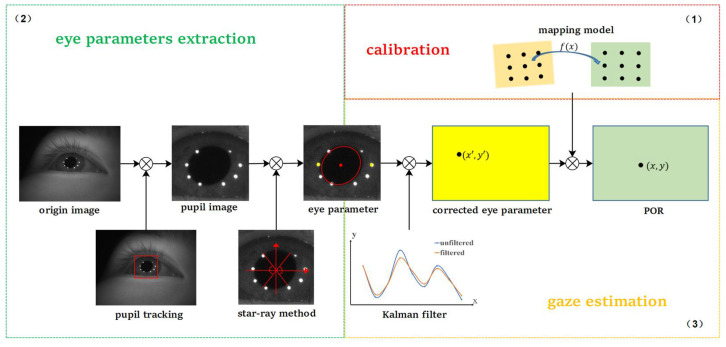
Flowchart of the proposed algorithm. (**1**) Schematic diagram of gaze tracking system calibration. (**2**) The process of eye parameters extraction. (**3**) The process of gaze estimation.

**Figure 2 sensors-22-03131-f002:**
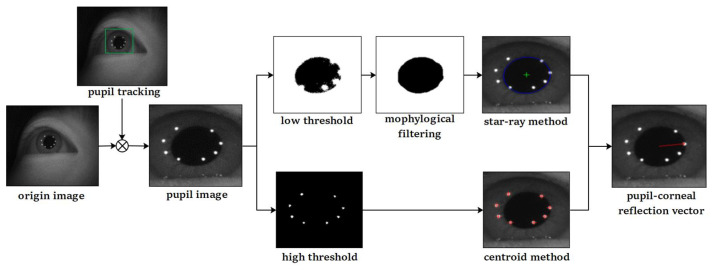
Pupil and Purkin spot-detection process.

**Figure 3 sensors-22-03131-f003:**
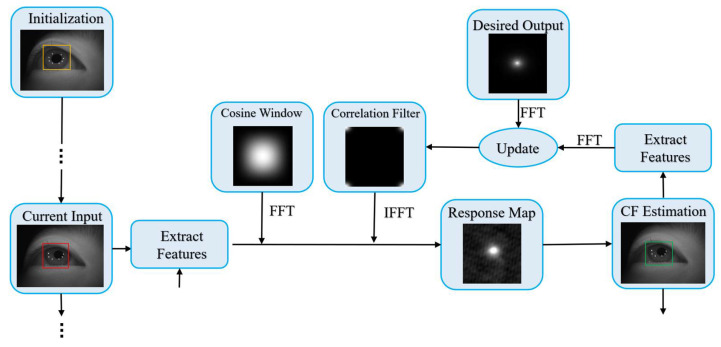
Flowchart of pupil tracking with KCF.

**Figure 4 sensors-22-03131-f004:**
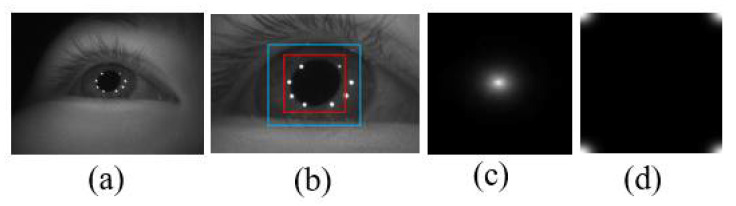
Initialization process. (**a**) The original image. (**b**) The image near the pupil; the red and blue rectangles represent the detected pupil image and the pupil image used for tracking, respectively. (**c**) A standard response map. (**d**) The initial correlation filter obtained from learning.

**Figure 5 sensors-22-03131-f005:**
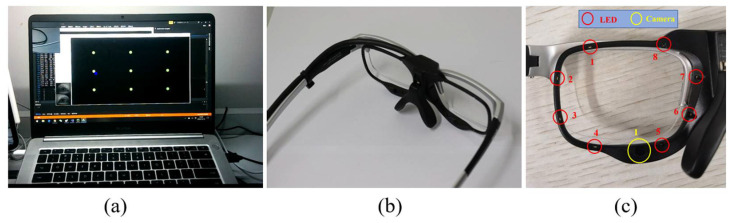
Equipment of gaze tracking system. (**a**) Laptop and tracking program. (**b**) aSee Glasses (Beijing 7INVENSUN Information Technology Co., Ltd., Beijing, China). (**c**) Schematic diagram of infrared LED lights and eye camera.

**Figure 6 sensors-22-03131-f006:**
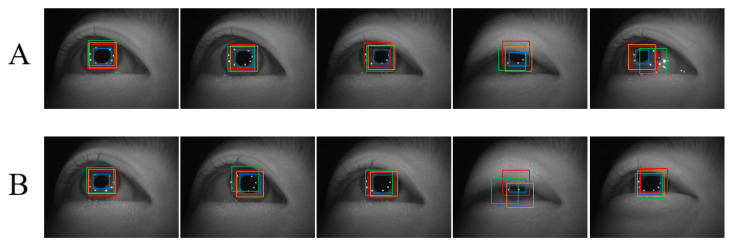
Results of pupil extraction. The blue, red, yellow, and green boxes represent the results of detection, MOSSE, KCF-gray, and KCF-hog, respectively. The upper figures are the results of pupil extraction in group (**A**), when the eyeballs moved randomly with normal blinking. The lower figures are the results of pupil extraction in group (**B**), when the eyeballs moved randomly with occasional squinting.

**Figure 7 sensors-22-03131-f007:**
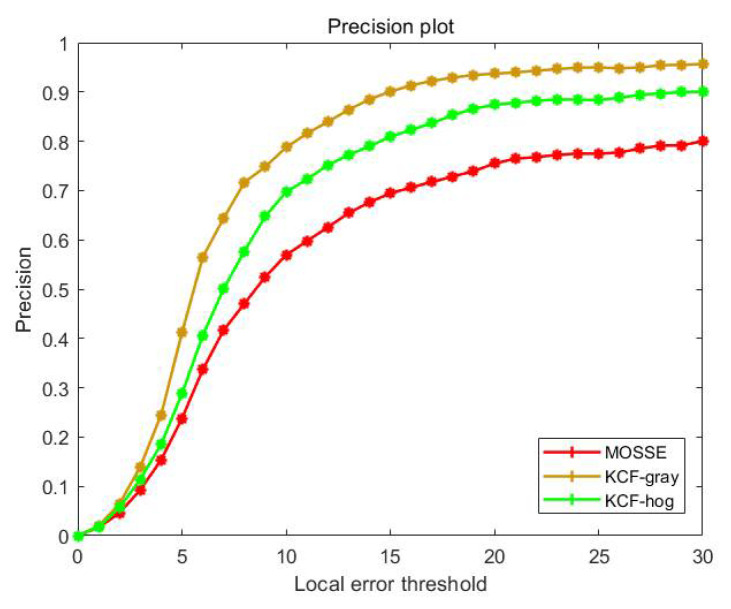
Precision plot for pupil tracking.

**Figure 8 sensors-22-03131-f008:**
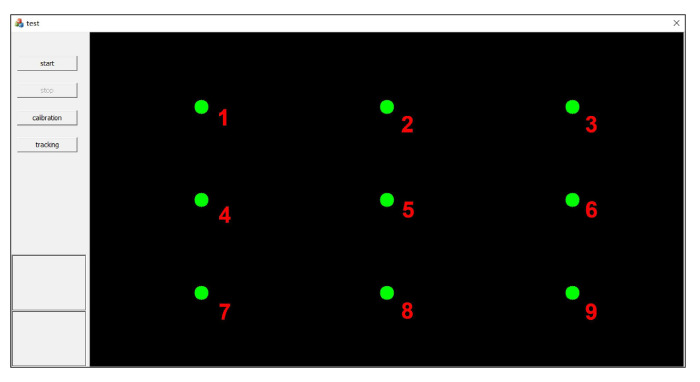
Numbered calibration points. The nine calibration points are sequentially numbered as 1 to 9.

**Figure 9 sensors-22-03131-f009:**
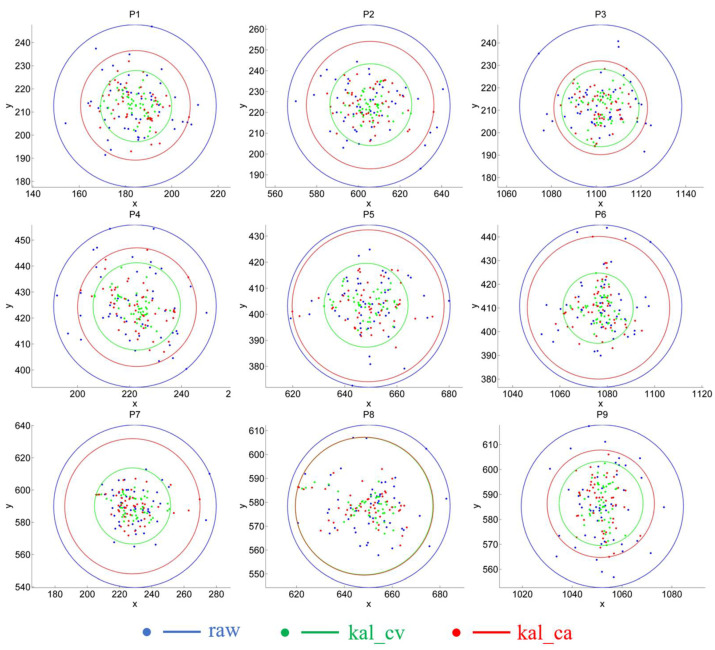
POR estimation of nine calibration points. (p1) to (p9) are the POR estimation results of points P1 to P9, respectively. The blue, green, and red points are the estimated PORs without filtering, with constant velocity model and with constant acceleration model respectively. The circles show the range of the PORs.

**Figure 10 sensors-22-03131-f010:**
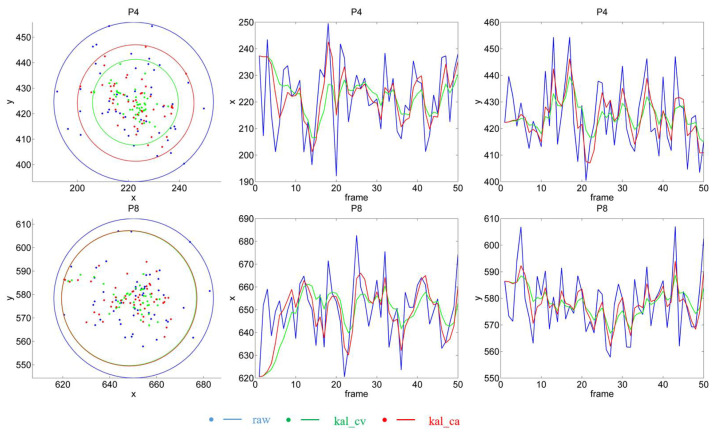
POR results of P4 and P8. The upper and lower rows present the results of P4 and P8, respectively. The left column is the actual POR, each dot represents a POR, and the circle represents the smallest covered circle, with the mean of the PORs as the center. The middle and right columns present the x- and y-values of each POR, respectively.

**Figure 11 sensors-22-03131-f011:**
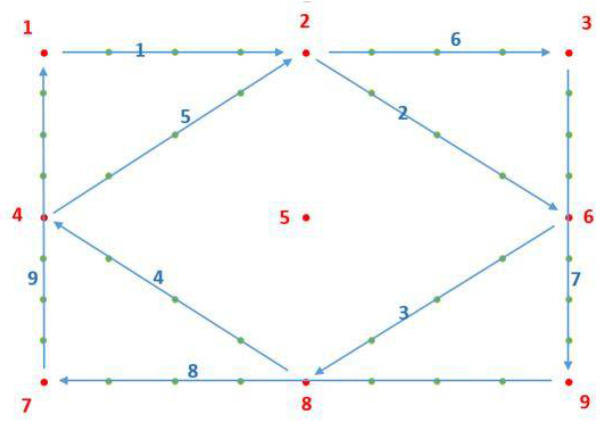
Turning points (with running number) and movement directions of the dot in the smooth pursuit task.

**Figure 12 sensors-22-03131-f012:**
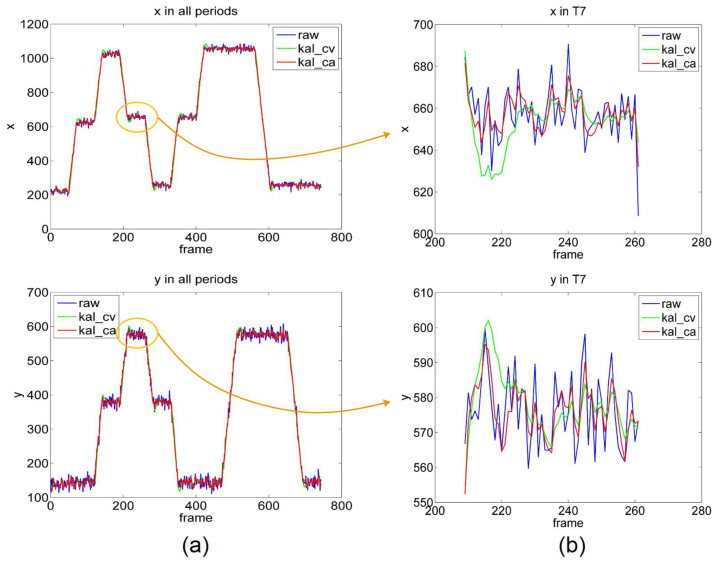
x- and y-values in the saccade task. (**a**) Overall result of all periods. (**b**) Local result of period T7.

**Figure 13 sensors-22-03131-f013:**
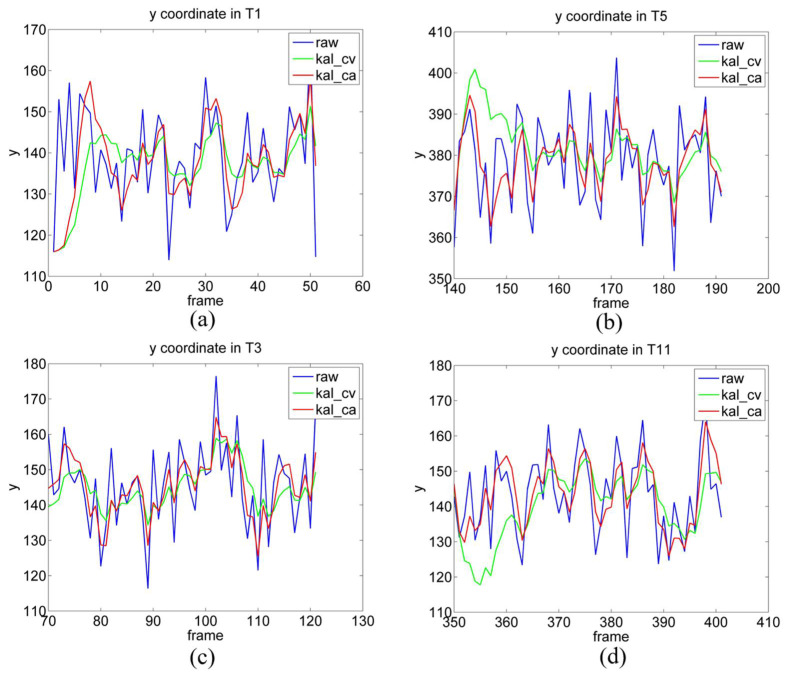
y-values in T1, T5, T3, and T11. (**a**–**d**) are the y-values of period T1, T5, T3 and T11, respectively.

**Figure 14 sensors-22-03131-f014:**
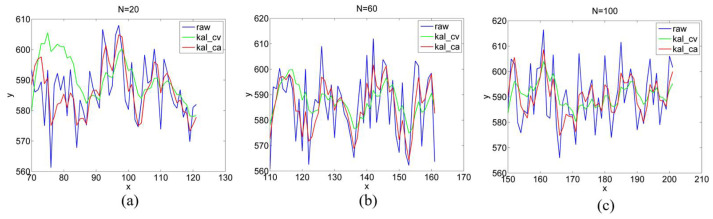
y-values of P8 with different numbers of motion images. (**a**–**c**) are y-values of P8 with 20, 60 and 100 motion images between P6 and P8.

**Figure 15 sensors-22-03131-f015:**
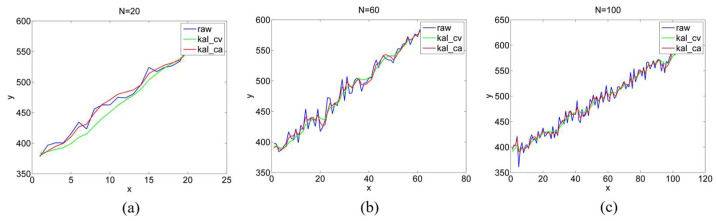
y-values between P6 and P8 with different numbers of motion images. (**a**–**c**) are y-values between P6 and P8 with 20, 60 and 100 motion images.

**Figure 16 sensors-22-03131-f016:**
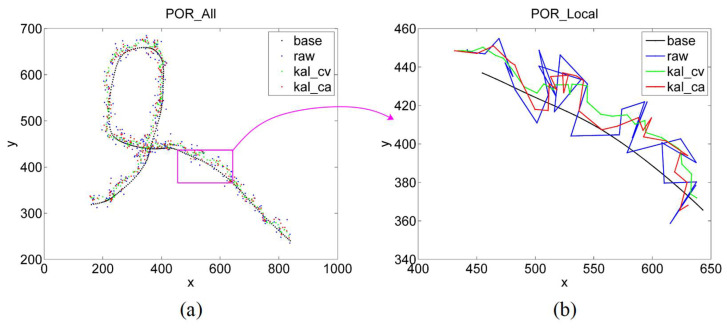
POR results of a randomly moving gaze point. (**a**) The all POR results. (**b**) A local POR results.

**Table 1 sensors-22-03131-t001:** Average speed and error of pupil tracking.

Metric	MOSSE	KCF-Gray	KCF-Hog
Speed	652	426	439
Average error	9.16	7.24	7.63

The unit of speed and average error are frames per second (FPS) and pixels, respectively.

**Table 2 sensors-22-03131-t002:** Mean error.

	P1	P2	P3	P4	P5	P6	P7	P8	P9
raw	77.67	**72.51**	81.62	68.16	45.49	63.85	34.51	25.00	30.30
kal_cv	**76.52**	72.53	**80.45**	67.04	**44.81**	**61.92**	**33.70**	22.73	**29.34**
kal_ca	77.18	72.92	80.75	**66.92**	45.32	62.45	34.40	23.44	29.68

The unit of mean error is pixel.

**Table 3 sensors-22-03131-t003:** Variance.

	P1	P2	P3	P4	P5	P6	P7	P8	P9
raw	149.03	172.57	134.26	183.11	112.07	159.41	136.62	119.94	156.30
kal_cv	**43.77**	**32.09**	**44.32**	**34.90**	**26.43**	**31.78**	**61.50**	**26.54**	**44.98**
kal_ca	98.16	81.33	58.83	88.21	55.73	78.43	82.09	43.55	69.55

The unit of variance is pixel^2^.

**Table 4 sensors-22-03131-t004:** Mean and variance of all points.

	2D Model		3D Model	
	Mean Error	Variance	Mean Error	Variance
raw	55.56	147.04	49.35	124.42
kal_cv	**54.34**	**38.48**	**48.57**	**37.53**
kal_ca	54.78	72.88	48.83	60.86

The unit of mean error and variance are pixel and pixel^2^, respectively.

**Table 5 sensors-22-03131-t005:** Description of the time periods.

Point	P1		P2		P6		P8		P4		P2		P3		P9		P7		P1
Period	T1	T2	T3	T4	T5	T6	T7	T8	T9	T10	T11	T12	T13	T14	T15	T16	T17	T18	T19

## Data Availability

The data presented in this study are available in the article.

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
