# Peer review of "Stable Gaze Tracking with Filtering Based on Internet of Things"

_sensors, 2022, doi:10.3390/s22093131_

Round 1

Reviewer 1 Report

The paper introuduce a new approach for improving the performance of gaze tracking in an active infrared source gaze tracking system. It is relatively well written, but with several parts need to be clearified, corrected detailed, as marked in the attached file.

Reviewer 2 Report

Comments to the Author

The topic of this study is interesting, but a major revision is required.

  1. In Section 3, several ideas and methods are not proposed by the authors. The authors should focus on the proposed original method. The ideas and methods that are not proposed by the authors should be presented in a background section.
  2. The authors should explain why they adopted Kalman filter (KF) for their applications. Besides, the implementation of KF is suggested to be deeply described.
  3. The authors are suggested to present what new ideas are proposed by them. The authors should present what contributions and advanced techniques are done by them. Because, the Gaze Tracking has become popular and widely applied.
  4. The authors should give mathematical models say how to optimize the KF. Moreover, the authors better to give the mathematical models to prove the performance of the KF.
  5. The authors are suggested to explain why they didn't adopt other suitable filters, such as Hopfield, and so on Bi-LSTM, RNN, GRU, Bi-RNN, and Bi-GRU.
  6. The authors are suggested to compare the original method with other methods.
  7. The authors are suggested to highlight the contributions of this study in the first section.
  8. The authors are suggested to survey and cite more recent publications that have been published in refereed journals in the recent three years.
  9. The limitation of the proposed method should be discussed in the last section.
  10. The future work of this study should be discussed in the last section. The challenges and difficulties should be introduced comprehensively.
  11. It is believed that the discussion of related work on of graze tracking can help to improve the depth of this paper.
  12. The hyper-parameters adopted by the KF should be reported. The Experiment for the process is encouraged to be presented.
  13. specifically, the motivation behind the proposed method is suggested to be introduced.

Round 2

Reviewer 2 Report

Comments to the sensors-1606066_V2

Though, most suggested places are polished carefully, there have some minor flaws are required to decorated.

  1. Check up all of captures is necessary. The enlarge to the graphs shown in some figures is suggested for improving the readability, for example, Figures 1 to 2, and so on.
  2. The authors are suggested to label the x-Axis and y-Axis shown in Figure 9.
  3. The authors are also suggested to present how to optimize the performance for the proposed methodologies in advance.
